

# Mining the potential prognostic value of synaptosomal-associated protein 25 (SNAP25) in colon cancer based on stromal-immune score

Jinyan Zou[1], Darong Duan[2], Changfa Yu[2], Jie Pan[3], Jinwei Xia[2], Zaixing Yang[2] and Shasha Cai[2]

[1] Department of Gastroenterology, Taizhou First People's Hospital, Huangyan Hospital of Wenzhou Medical University, Zhejiang, China, Taizhou, Zhejiang, China

[2] Department of Laboratory Medicine, Taizhou First People's Hospital, Huangyan Hospital of Wenzhou Medical University, Zhejiang, China, Taizhou, Zhejiang, China

[3] Outpatient Department, Taizhou First People's Hospital, Huangyan Hospital of Wenzhou Medical University, Zhejiang, China, Taizhou, Zhejiang, China

Corresponding author
Shasha Cai, tyycss@163.com

## ABSTRACT

**Background**. Colon cancer is one of the deadliest tumors worldwide. Stromal cells and immune cells play important roles in cancer biology and microenvironment across different types of cancer. This study aimed to identify the prognostic value of stromal/immune cell-associated genes for colon cancer in The Cancer Genome Atlas (TCGA) database using bioinformatic technology.

**Methods**. The gene expression data and corresponding clinical information of colon cancer were downloaded from TCGA database. Stromal and immune scores were estimated based on the ESTIMATE algorithm. Sanger software was used to identify the differentially expressed genes (DEGs) and prognostic DEGs based on stromal and immune scores. External validation of prognostic biomarkers was conducted in Gene Expression Omnibus (GEO) database. Gene ontology (GO) analysis, pathway enrichment analysis, and gene set enrichment analysis (GSEA) were used for functional analysis. STRING and Cytoscape were used to assess the protein-protein interaction (PPI) network and screen hub genes. Quantitative real-time PCR (qRT-PCR) was used to validate the expression of hub genes in clinical tissues. Synaptosomal-associated protein 25 (SNAP25) was selected for analyzing its correlations with tumor-immune system in the TISIDB database.

**Results**. Worse overall survivals of colon cancer patients were found in high stromal score group (2963 vs. 1930 days, log-rank test $P = 0.038$) and high immune score group (2894 vs. 2230 days, log-rank test $P = 0.076$). 563 up-regulated and 9 down-regulated genes were identified as stromal-immune score-related DEGs. 70 up-regulated DEGs associated with poor outcomes were identified by COX proportional hazard regression model, and 15 hub genes were selected later. Then, we verified aquaporin 4 (AQP4) and SNAP25 as prognostic biomarkers in GEO database. qRT-PCR results revealed that AQP4 and SNAP25 were significantly elevated in colon cancer tissues compared with adjacent normal tissues ($P = 0.003, 0.001$). GSEA and TISIDB suggested that SNAP25 involved in cancer-related signaling pathway, immunity and metabolism progresses.

**Conclusion**. SNAP25 is a microenvironment-related and immune-related gene that can predict poor outcomes in colon cancer.

## INTRODUCTION

Colon cancer is the third most common malignant tumors and one of the leading causes of cancer-related death worldwide. Since risk factors have been investigated, earlier detection, cancer prevention, surgical techniques, radiotherapy and chemotherapy treatment have been improved, the incidence and mortality of colon cancer has slowly declined. A model-based estimate showed that 104,610 new cases of colon cancer would be diagnosed and 53,200 patients would die of colon cancer in United States (*Siegel, Miller & Jemal, 2020*). However, the incidence of colorectal cancer in patients aged <50 years increased by 22% from 2000–2013 (*Siegel et al., 2017*). Thus, more attention should be focused on early diagnosis through screening and accurately predicting the survival outcome of patients with colon cancer.

Stromal cells and immune cells form the major fraction of colon cancer tissue and are associated with tumor progress, inflammatory and metabolic disorders (*Ghesquière et al., 2014*; *Nilendu et al., 2018*). An increasing amount of studies have highlighted the importance of stromal cells and immune cells in cancer biology and microenvironment across different types of cancers (*Barros Jr et al., 2018*; *Garcia-Gomez, Rodríguez-Ubreva & Ballestar, 2018*; *Zhan et al., 2017*). ESTIMATE is a new method that infers stromal and immune cells in malignant tumors using gene expression signatures (*Yoshihara et al., 2013*), and has been conducted in acute myeloid leukemia (*Yan et al., 2019*), gastric cancer (*Wang, Wu & Chen, 2019*), and glioblastoma (*Jia et al., 2018*).

In the current study, we obtained immune and stromal scores of colon cancer based on ESTIMATE. To help elucidate the stromal-immune score-based genes with prognostic value in colon cancer, we obtained gene expression dates from The Cancer Genome Atlas (TCGA), and verified the survival value in a different colon cancer cohort available from the Gene Expression Omnibus (GEO) database. Two hub genes were validated to be prognosis biomarkers and selected for further analysis. We investigated the potential underlying mechanisms of synaptosomal-associated protein 25 (SNAP25) in cancer-related signaling pathway, immunity and metabolism progresses through gene set enrichment analysis (GSEA) and TISIDB database.

## MATERIALS & METHODS

### Database and estimation of stromal and immune scores

The TCGA level 3 gene expression data and corresponding clinical information for colon cancer patients were obtained from Genomic Date Commons of the National Cancer Institute (http://portal.gdc.cancer.gov). Only patients with gene expression data, follow up information and clinicopathologic information were included in this study. For normalization, the RNA-seq data of all patients was transformed to transcripts per million (TPM) values (https://pubmed.ncbi.nlm.nih.gov/30379987/). The stromal and immune

scores for each TCGA sample were conducted by R 3.6.2 using the R package "estimate" (https://pubmed.ncbi.nlm.nih.gov/24113773/).

We obtained gene expression profiles and clinical information of 430 colon cancer patients from TCGA database. Among them, 231 (53.7%) cases were male and 199 (46.3%) cases were female. The average age of patients at initial pathological diagnosis was 66.3 years (range: 31-90 years). Histologic diagnosis included 369 (85.5%) cases of colon adenocarcinoma and 57 (13.3%) cases of colon mucinous adenocarcinoma, 4 (0.9%) cases were not classified. The tumor stage was stage I in 17.4%, stage II in 38.6%, stage III in 29.1, stage IV in 14.4% of cases, and 2 cases (0.5%) were of unknown stage. The tumors were located in the left (40%) or right (55.3%) colon according to their anatomic neoplasm subdivision, with 4.7% unknown. Based on ESTIMATE algorithm, we obtained stromal scores (range: $-2262.07 \sim 1999.52$) and immune scores (range: $-954.97 \sim 3035.59$) for all these colon cancer patients.

## Correlations between clinicopathologic data and stromal/immune score

The correlations between clinicopathologic data and stromal/immune score were analyzed by SPSS 22.0 software (SPSS, Inc., Chicago, IL, USA). Patients with colon cancer were divided into high stromal/immune score (the fourth quartile) and low stromal/immune score groups (quartile 1–3). The stromal/immune score of different clinicopathologic groups was compared by Mann–Whitney U test, and overall survival was estimated by the Kaplan–Meier method and compared by log-rank tests. A value of $P < 0.05$ was considered statistically significant.

## Identification of differentially expressed genes (DEGs)

DEGs were identified based on immune and stromal scores (high stromal/immune score group vs. low stromal/immune score group) by Sanger_V1.0.8 software (https://shengxin.ren/softs/Sanger_V1.0.8.zip). Genes with log2 (fold change) >1.5 or < -1 combined with a $P$ value < 0.01 were defined as DEGs. The volcano plot of the DEGs was drawn by Sanger_V1.0.8 software, and the venn diagram was drawn on a website Venny 2.1.0 (https://bioinfogp.cnb.csic.es/tools/venny/index.html).

## Gene ontology (GO) and pathway enrichment analyses

Cellular component (CC), molecular function (MF), biological process (BP) and pathway enrichment analyses were conducted using FunRich 3.1.3 (https://pubmed.ncbi.nlm.nih.gov/25921073/). A value of $P < 0.05$ was considered as the screening condition.

## Survival analysis

A COX proportional hazards model was applied to illuminate prognostic DEGs of colon cancer obtained from TCGA and a $P$ value < 0.01 was considered significant. An open source web tool PrognoScan (https://pubmed.ncbi.nlm.nih.gov/19393097/) was conducted to verify the survival outcomes between prognostic DEGs identified and colon cancer patients from GEO database.

## Protein–protein interaction (PPI) network construction

The PPI network for DEG-encoded proteins was performed by STRING database (https://pubmed.ncbi.nlm.nih.gov/30476243/) and reconstructed by Cytoscape 3.7.2 (https://pubmed.ncbi.nlm.nih.gov/14597658/). The most significant modular analysis was identified by Molecular Complex Detection (MCODE) plugin of Cytoscape, and the plug-in Biological Networks Gene Oncology (BiNGO) of Cytoscape was applied to analysis GO term of hub genes.

## Heatmap and clustering analysis

Heatmap and clustering analysis were completed by "heatmap" package.

## Quantitative real-time PCR (qRT-PCR)

Total RNA was extracted with Trizol reagent (TaKaRa Bio Inc. Shiga, Japan) from colon cancer and adjacent normal tissues. cDNA was synthesized with PrimeScript$^{TM}$ RT reagent kit (TaKaRa, RR036A). Quantitative real-time PCR (qRT-PCR) was carried out using the TB Green$^{TM}$ Premix Ex Taq$^{TM}$ kit (TaKaRa, RR420A) on ABI step one Real-Time PCR system. The primers were as follows: AQP4, sense strand 5′- GAGCAGGAATCCTCTATC-3′, antisense strand 5′- AGTGACATCAGTCCGTTT-3′; SNAP25, sense strand 5′-GTAGTGGACGAACGGGAGC-3′, antisense strand 5′- CCTGTCGATCTGGCGATT-3′; GAPDH, sense strand 5′-GTCAACGGATTTGGTCTGTATT-3′, antisense strand 5′-AGTCTTCTGGGTGGCAGTGAT-3′.

The Institutional Medical Ethics Review Board of Taizhou first people's hospital in Zhejiang Province approval to carry out the study within its facilities (Ethical Application Ref: 2019-KY009-03).

## Gene set enrichment analysis (GSEA)

According to the expression level of AQP4 (or SNAP25), samples of the complete cohort from TCGA were divided into 2 groups, and implemented using GSEA by Sanger_V1.0.8 software. The KEGG gene set biological process database (c2.cp.kegg.v6) were chosen for enrichment analysis. Terms with both $P$ value $< 0.01$ and false discovery rate (FDR) $< 0.01$ were identified.

## Mining the immune-related mechanism of SNAP25

TISIDB (https://pubmed.ncbi.nlm.nih.gov/14597658/) is a friendly web portal integrates 988 immune-related anti-tumor genes from 7 databases (*Ru et al., 2019*). The correlations between immune features and any gene may be explored in 30 TCGA cancer types. In this study, the TISIDB database was used to investigate the associations between expression (or methylation) of SNAP25 and tumor-infiltrating lymphocytes (TILs). However, the relations between TILs and expression (or methylation) of AQP4 in colon cancer were not integrated in TISIDB database. A value of $P < 0.05$ was considered statistically significant.

# RESULTS

## The relationships between stromal/immune score and clinical features

The stromal and immune scores were variously distributed between adenocarcinoma and mucinous adenocarcinoma (Figs. 1A, 1C). Both stromal score and immune score of colon

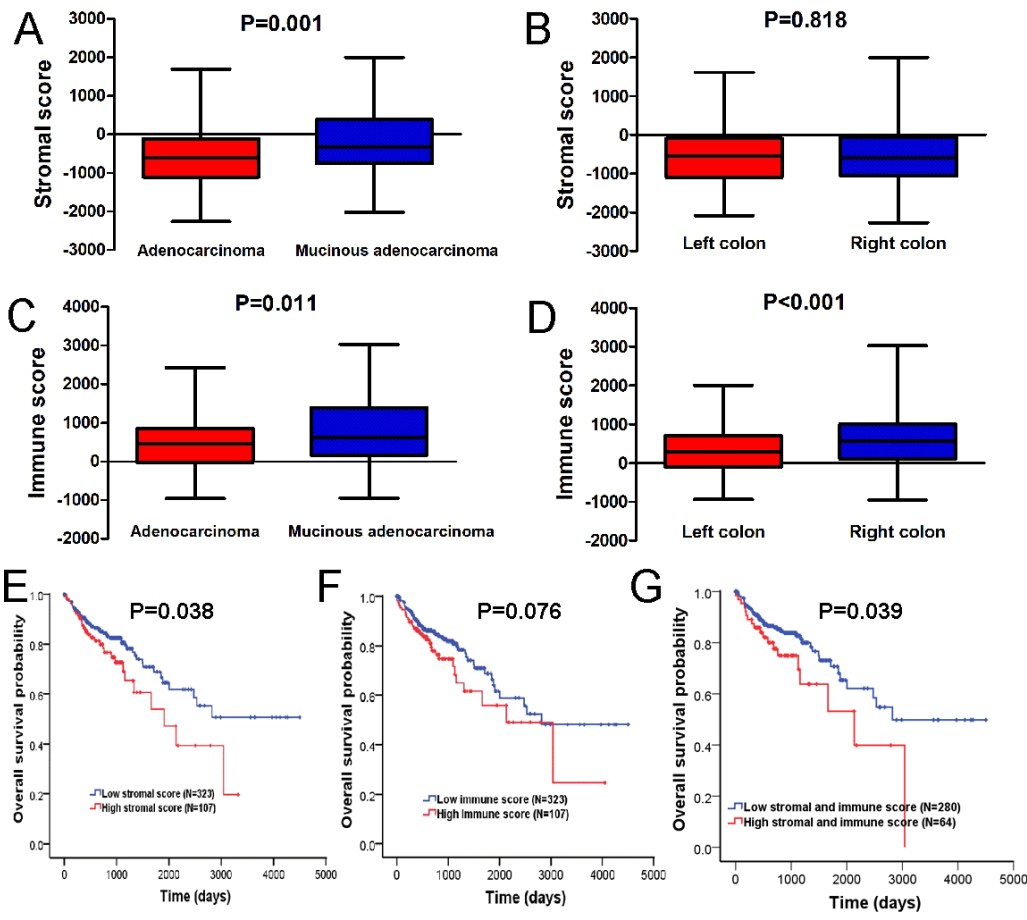

**Figure 1** **Stromal and immune scores are associated with the clinical features of colon cancer and their overall survival.** (A) Distribution of stromal scores of colon cancer between different histologic diagnosis. (B) The correlation between stromal scores and anatomic neoplasm subdivision. (C) Distribution of immune scores of colon cancer between different histologic diagnosis. (D) The correlation between immune scores and anatomic neoplasm subdivision. (E) Kaplan–Meier survival curve for patients with low vs. high stromal scores. (F) Kaplan–Meier survival curve for patients with low vs. high immune scores. (G) Kaplan–Meier survival curve for patients with low vs. high both stromal and immune scores.

adenocarcinoma cases were significantly lower than those of mucinous adenocarcinoma cases ($P = 0.001$, $0.011$, respectively). In addition, right colon tumors yielded higher immune scores than those left colon cases (Fig. 1D, $P < 0.001$), though no significant differences between left and right colon were found for the stromal scores (Fig. 1B, $P = 0.818$).

The potential association of overall survival and stromal/immune score was explored by classifying 430 colon cancer patients into high and low score groups based on their stromal or immune scores. As shown in Figs. 1E, 1F, the median overall survival of patients with a low stromal score was longer than those in high score group (2963 vs. 1930 days, log-rank test $P = 0.038$); consistently, the median overall survival of patients with a low immune score was longer than those in high score group (2894 vs. 2230 days, log-rank

test $P = 0.076$), although there was no statistically significant difference. However, patients with both a high stromal score and a high immune score were found to have significantly worse survival than those with low scores (1891 vs. 2974 days, log-rank test $P = 0.039$) (Fig. 1G).

### Identification of DEGs by stromal and immune scores in colon cancer

After standardization of the RNA-Seq data for all 430 colon cancer patients obtained from TCGA database, we identified 4881 and 1512 DEGs based on stromal and immune scores, respectively. As shown in the volcano plots of DEGs for stromal/immune score (Figs. 2A, 2B), 4,791 genes were up-regulated and 90 genes were down-regulated for the comparison based on stromal score. Similarly, 1113 genes were up-regulated and 399 genes were down-regulated based on high immune score group vs. low immune score group. Through Venn diagram (Figs. 2C, 2D) analysis, 563 shared up-regulated DEGs and 9 shared down-regulated DEGs from stromal score and immune score groups were identified and selected for subsequent analysis.

### Functional and pathway enrichment analyses

Go and pathway enrichment analyses of the above 572 genes were performed (Figs. 2E, 2F, 2G, 2H). For CC, DEGs were mainly associated with plasma membrane, extracellular and extracellular space. With regard to MF, genes were mainly clustered in receptor activity, cell adhesion molecule activity and B cell receptor activity. DEGs in the BP category primarily enriched in immune response, cell communication and signal transduction. The pathway enrichment analysis showed genes were mainly enriched in epithelial-to-mesenchymal transition, peptide ligand-binding receptors and GPCR ligand binding.

### Identification of prognostic DEGs in colon caner

The COX proportional hazard regression model was constructed to identify potential prognostic DEGs in colon cancer. Among the 563 shared up-regulated DEGs and 9 shared down-regulated DEGs, 70 up-regulated DEGs associated with poor outcomes were shown in Fig. 3.

### PPI network construction and hub gene analysis among prognostic DEGs

To further explore the interplay among the 70 identified prognostic DEGs, we conducted a PPI network containing 30 nodes and 53 edges based on STRING tool and Cytoscape software (Fig. 4A). Module analysis using MCODE was constructed, and 15 hub genes were selected (Fig. 4A). CC, MF, BP analyses of the total 15 hub genes were performed using BiNGO (Fig. 4B). The 15 hub genes were mainly associated with plasma membrane, transporter activity, secretion and channel activity.

### Heatmap and clustering analysis of 15 hub genes

The expression level of 15 hub genes in "dead" and "alive" groups was shown in Fig. 4C.

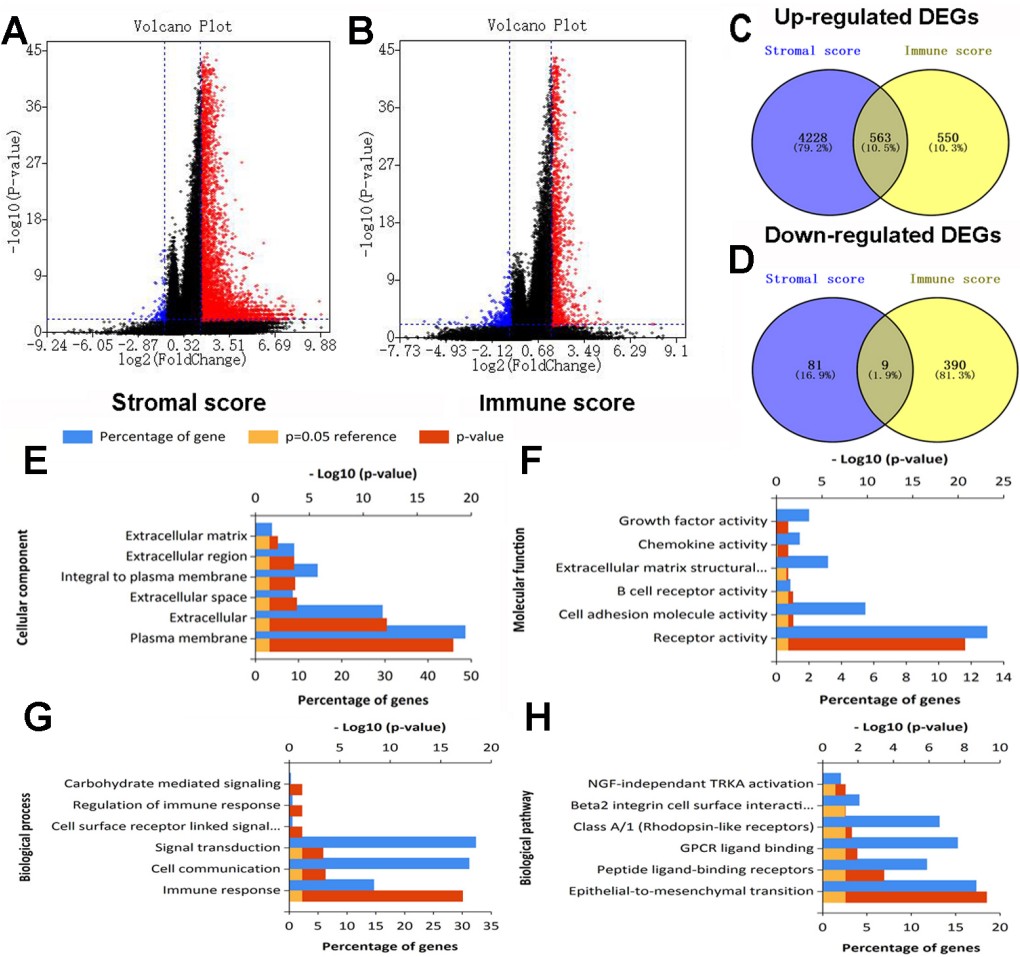

**Figure 2** Expression profiles and biological functions of DEGs based on stromal and immune scores. (A) Volcano plot showing up-regulated DEGs in red and down-regulated DEGs in blue for the comparison based on high and low stromal score groups. (B) Volcano plot showing up-regulated DEGs in red and down-regulated DEGs in green for the comparison based on high and low immune score groups. (C, D) Venn diagrams showing 563 shared up-regulated DEGs (C) and nine shared down-regulated DEGs (D) from stromal score and immune score groups. (E, F, G, H) Top six Go terms and pathways enriched by DEGs.

## Verifying the survival outcomes of hub genes in the GEO database

The survival outcomes of 15 hub genes were identified on the PrognoScan web tool, which provided overall survival of GSE12945, GSE17536 and GSE17537 datasets for colorectal cancer. Then, AQP4 and SNAP25 were verified (Fig. 5) to be significantly associated with overall survival according to both the log-rank test and COX proportional hazards regression analysis (all $P < 0.05$).

## The expression of two hub genes in 20 colon cancer and adjacent normal tissue sample set using qRT-PCR

The expression of AQP4 and SNAP25 was validated in 20 pairs of clinical tissues using qRT-PCR. Interestingly, the mRNA relative expression levels of both AQP4 and SNAP25

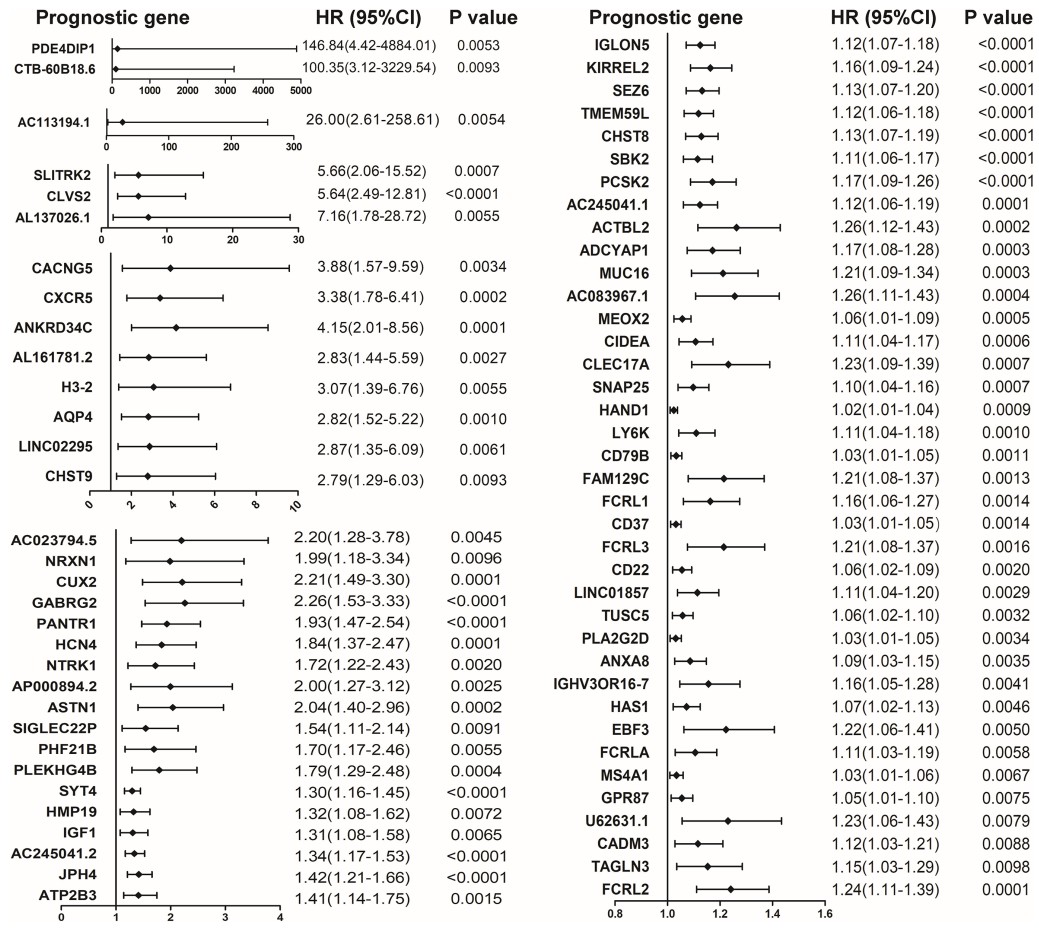

**Figure 3** Forest plot of hazard ratios (HR) for 70 prognostic DEGs in colon cancer. HR and 95% confidence intervals (CI) were obtained by the COX proportional hazards model. HR and 95% confidence intervals (CI) were obtained by the COX proportional hazards model.

were significantly elevated in colon cancer tissues compared with adjacent normal tissues ($P = 0.003$, $0.001$) (Fig. 6).

## GSEA using TCGA database

To further investigate the underlying mechanism of AQP4 and SNAP25 in colon cancer, KEGG pathway enrichment analysis was performed by GSEA. For AQP4, "vascular smooth muscle contraction" (NES = 2.17, $P < 0.001$, FDR = 0.006) gene set was prominently enriched. For SNAP25, 57 gene sets were enriched, including 11 gene sets which were cancer-related processes (Fig. 7). Besides, high expression of SNAP25 might also be involved in "B cell receptor signaling pathway" (NES = 2.06, $P = 0.002$, FDR = 0.002), "cell adhesion molecules cams" (NES = 2.05, $P = 0.008$, FDR = 0.002), "chemokine signaling pathway" (NES = 2.07, $P = 0.002$, FDR = 0.002), "complement and coagulation cascades" (NES = 2.08, $P < 0.001$, FDR = 0.002), "T cell receptor signaling pathway" (NES = 2.06, $P < 0.001$, FDR = 0.002), "adipocytokine signaling pathway" (NES = 1.96,

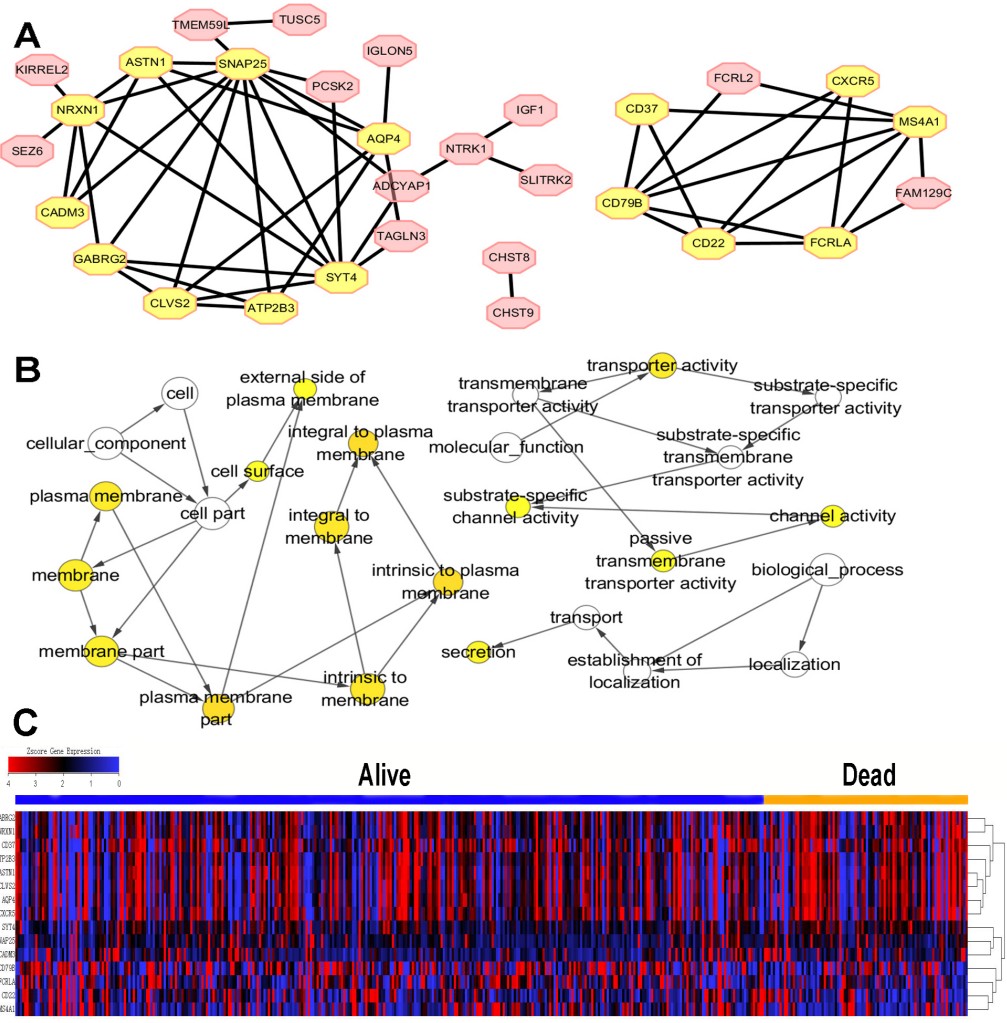

**Figure 4 PPI network, GO analysis and expression of hub gene.** (A) PPI network contained 30 nodes and 53 edges based on STRING tool and Cytoscape software was constructed. The hub genes were selected by MCODE in yellow with 9 nodes and 22 edges in the left module, and six nodes and 12 edges in the right module. (B) The GO analysis of hub genes was performed using BiNGO. The color of the node represented the corrected $P$-value of ontologies, and the size of the node represented the numbers of involved genes in the ontologies. $P < 0.01$ was considered statistically significant. (C) The expression of 15 hub genes in TCGA cohort by $z$-score, with red represents higher expression and blue represents lower expression.

$P < 0.001$, FDR $= 0.005$), "aldosterone regulated sodium reabsorption" (NES $= 2.02$, $P < 0.001$, FDR $= 0.003$), "glycosaminoglycan biosynthesis heparan sulfate" (NES $= 2.00$, $P < 0.001$, FDR $= 0.004$), and "insulin signaling pathway" (NES $= 1.94$, $P = 0.002$, FDR $= 0.007$). This suggested that immunity and metabolism may be as well involved in the underlying mechanism of SNAP25 in colon cancer.

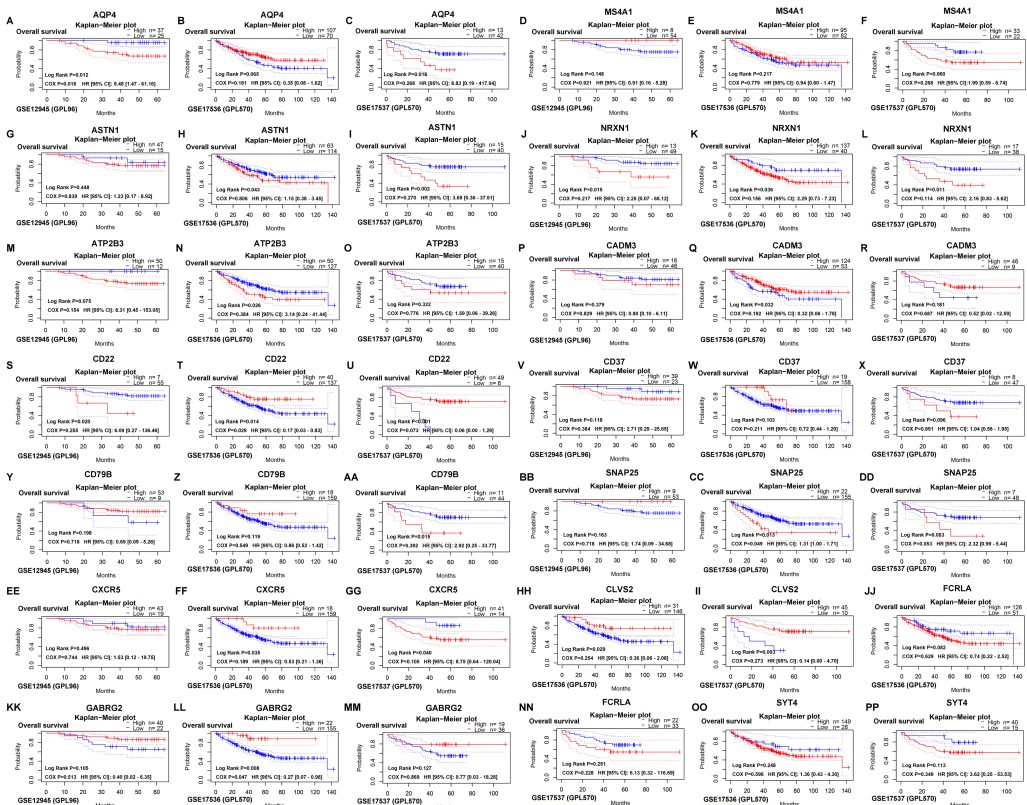

**Figure 5** Verifying the survival outcomes of hub genes including AQP4 (A, B, C), MS4A1 (D, E, F), ASTN1 (G, H, I), NRXN1 (J, K, L), ATP2B3 (M, N, O), CADM3 (P, Q, R), CD22 (S, T, U), CD37 (V, W, X), CD79B (Y, Z, AA), SNAP25 (BB, CC, DD), CXCR5 (EE, FF, GG), CLVS2 (HH, II), FCRLA (JJ, NN), GABRG2 (KK, LL, MM), SYT4 (OO, PP) in the GEO database. Kaplan–Meier survival curves with the log-rank test and COX proportional hazards regression analysis were generated for prognostic genes. The dotted blue/ red lines indicated 95% confidence intervals of overall survival probability. $P < 0.05$ was considered statistically significant.

## Regulation of immune molecules by SNAP25

The spearman's correlations between lymphocytes and expression, methylation of SNAP25 were performed in TISIDB database (Fig. 8). The associations between the expression of SNAP25 and immune-related signatures of TILs types were shown in Fig. 8A, and the greatest correlations including natural killer cell (NK; $r = 0.493$, $P < 2.2e{-}16$), macrophage ($r = 0.45$, $P < 2.2e{-}16$), mast cell ($r = 0.448$, $P < 2.2e{-}16$), and natural killer T cell (NKT; $r = 0.447$, $P < 2.2e{-}16$) were shown in Fig. 8B. The correlations between methylation of SNAP25 and lymphocytes were shown in Fig. 8C, and Fig. 8D displayed the remarkable negative correlations including plasmacytoid dendritic cell (pDC; $r = -0.419$, $P = 2.96e{-}14$), type 1 T helper cell (Th1; $r = -0.406$, $P = 4.06e{-}13$), T follicular helper cell (Tfh; $r = -0.391$, $P = 3.78e{-}12$), and NKT ($r = -0.391$, $P = 3.61e{-}12$). Therefore, the potential underlying mechanism of SNAP25 in colon cancer may be involved in the regulation of the above TILs.

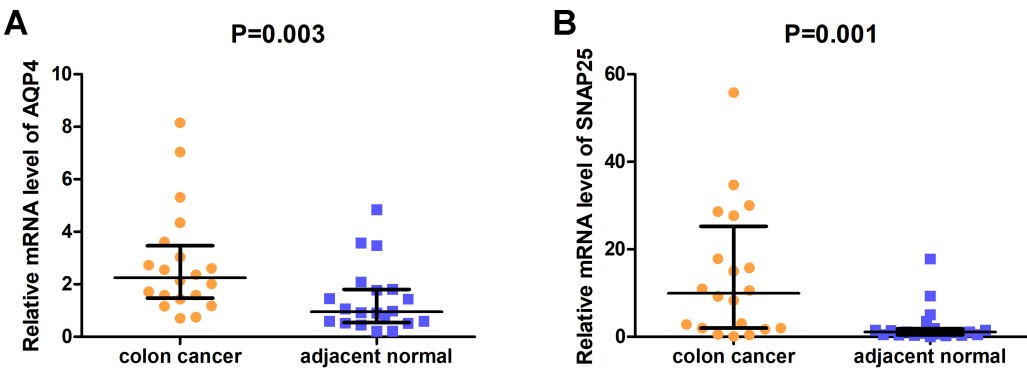

**Figure 6** The relative mRNA expression levels of AQP4 (A) and SNAP25 (B) in colon cancer tissues and adjacent normal tissues were confirmed by qRT-PCR.

## DISCUSSION

Solid tumor tissues comprise not only malignant cells but also tumor microenvironment, including immune cells, stromal cells, epithelial cells, fibroblasts, vascular cells and signaling molecules. Accumulating evidence clarifies that tumor microenvironment plays a crucial role in tumor growth, progression, metastasis, prognosis, and treatment (*Petitprez et al., 2018*; *Wu & Dai, 2017*). In the current study, we focused on stromal and immune scores, which reflect the microenvironment of tumor and hence contribute to survival prediction in colon cancer. Meanwhile, our results were in accordance with previous specific insights (*Koi & Carethers, 2017*; *Zhang et al., 2018*), and might provide extra data in the mining of interaction between tumor and environment in colon cancer.

Then, we screened out 572 microenvironment-related DEGs, and found they were mainly enriched in plasma membrane (CC), receptor activity (MF), immune response (BP), and epithelial-to-mesenchymal transition (pathway). Afterwards, AQP4, ASTN1, ATP2B3, CADM3, CD22, CD37, CD79B, CLVS2, CXCR5, FCRLA, GABRG2, MS4A1, NRXN1, SNAP25, and SYT4 were identified as prognostic hub genes, and two of them were verified to be prognosis biomarkers in GEO database. qRT-PCR results revealed that AQP4 and SNAP25 were significantly elevated in colon cancer tissues compared with adjacent normal tissues ($P = 0.003, 0.001$). Next, we investigated the underlying mechanism of AQP4 and SNAP25 in colon cancer by GSEA, and found that the high expression of SNAP25 might be involved in cancer-related signaling pathway, immunity and metabolism processes. Further researches in TISIDB database indicated greatest positive correlations between SNAP25 expression and TILs (NK, NKT, macrophage, mast), and greatest negative correlations between SNAP25 methylation and TILs (pDC, Th1, Tfh, NKT). TILs are associated with prognosis for the survival of various tumors in many previous studies. TILs are also reported as a predictive biomarker in colon cancer (*Zhao et al., 2019*). However, some studies found no significant association between TILs and overall survival. It remains controversial on the prognostic value of TILs in colon cancer may be due to different TIL responses or subsets,

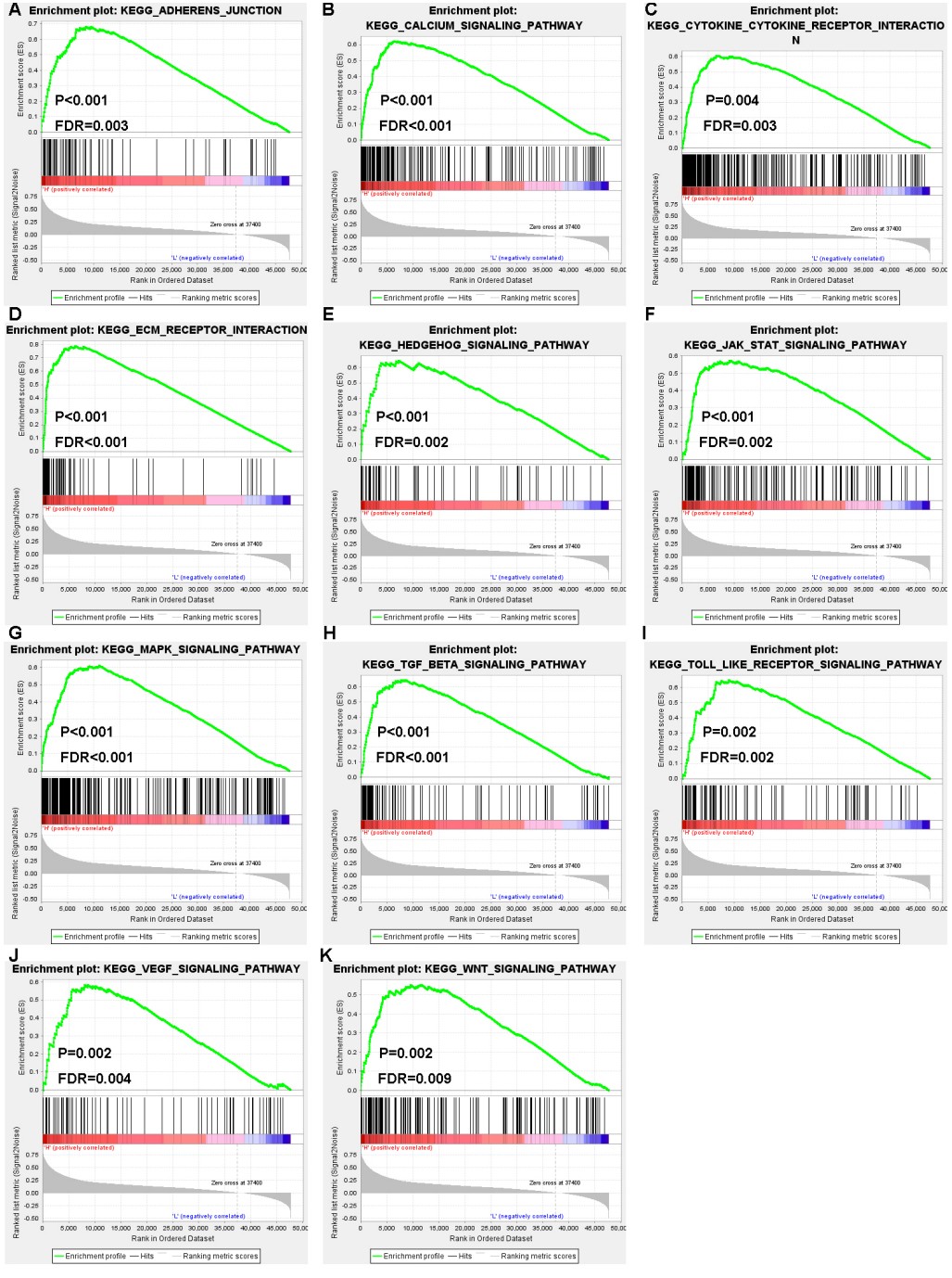

**Figure 7** **GSEA enrichment plots showed that eleven gene sets related to tumor signaling pathways (adherens junction (A), calcium signaling pathway (B), cytokine-cytokine receptor interaction (C), ECM receptor interaction (D), hedgehog signaling pathway (E), JAK–STAT signaling pathway (F), MAPK signaling pathway (G), TGF beta signaling pathway (H), Toll–like receptor signaling pathway (I), VEGF signaling pathway (J), Wnt signaling pathway (K)) were enriched in the high SNAP25 expression group.**

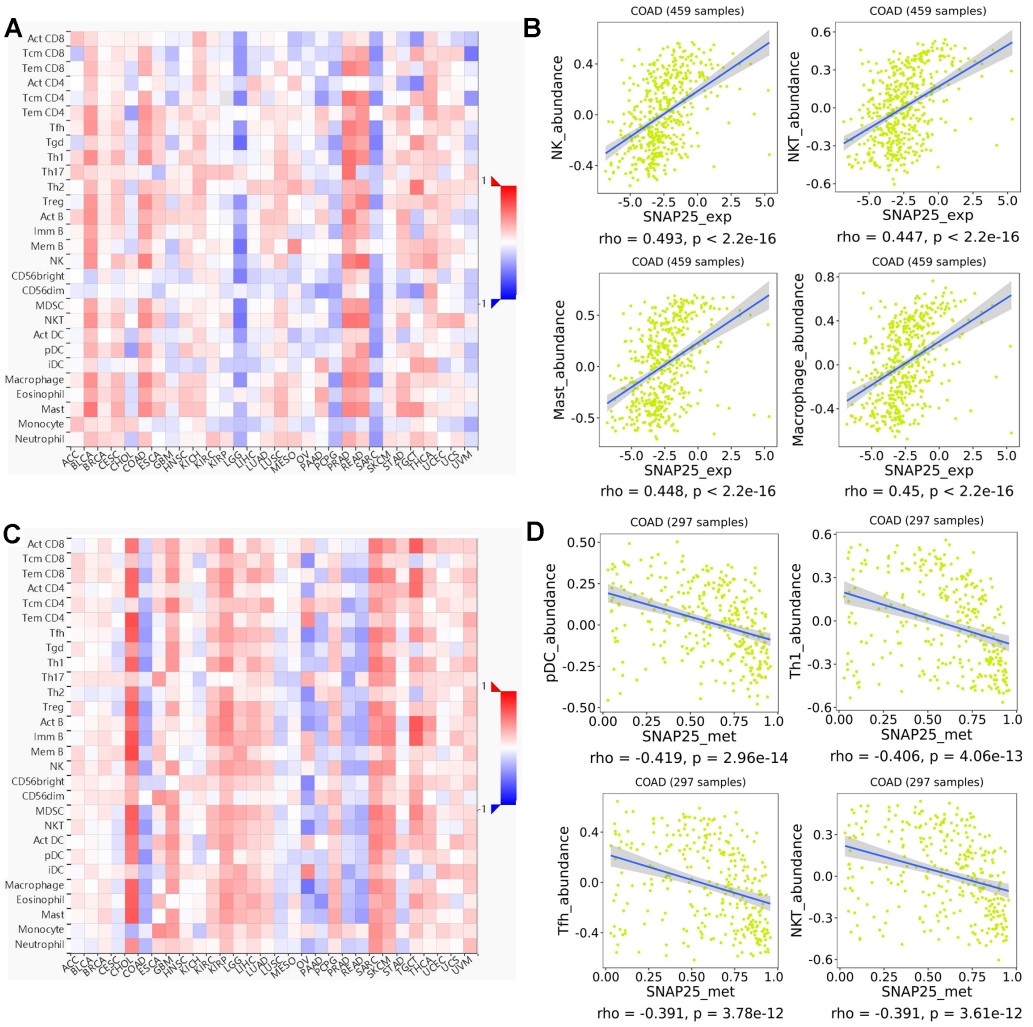

**Figure 8** **Spearman's correlations between SNAP25 and lymphocytes (TISIDB).** (A) Relations between the SNAP25 expression and abundance of TILs across human cancers. (B) Top four greatest positive correlations between SNAP25 expression and TILs. (C) Relations between the SNAP25 methylation and abundance of TILs across human cancers. (D) Top four greatest negative correlations between SNAP25 methylation and TILs.

different patient populations, different study designs, and the heterogeneity of immune infiltrate. Further studies should be designed to provide more insight into the study of TILs and survival.

SNAP25 belongs to the soluble N-ethylmaleimide-sensitive factor attachment protein receptor (SNARE) complex, which is essential for neurotransmitter release, synaptic, secretory vesicle exocytosis, minimal fusion machinery, cell-to-cell signaling, and the regulation of ion channels (*Baker & Hughson, 2016*; *Wang, Li & Hong, 2017*; *Yoon & Munson, 2018*). As reported, SNAP25 is potentially important for normal vesicle fusion and lysosomal trafficking (*Manca et al., 2014*; *Mu et al., 2018*). *Kobayashi et al. (2016)* found that SNAP25 protein was expressed in 46% (77/168) of diffuse large B-cell lymphoma

(DLBCL) patients and associated with CD5 expression ($P = 0.018$). *Huang et al. (2017)* displayed the expression and functional significance of SNAP25 in medulloblastoma.

Although the clinical significances and functions of SNAP25 for colon cancer have not been previously reported, it may serve as prognosis biomarker according to this study. To our greatest interest, SNAP25 is the most highly interconnected nodes (Fig. 4A), and involves in many cancer-related processes consists of adherens junction, calcium signaling pathway, cytokine receptor interaction, ECM receptor interaction, hedgehog signaling pathway, JAK-STAT signaling pathway, MAPK signaling pathway, TGF beta signaling pathway, Toll-like receptor signaling pathway, VEGF signaling pathway, and WNT signaling pathway (Fig. 7). Besides, SNAP25 is also involved in immunity and metabolism processes such as B cell receptor signaling pathway, cell adhesion molecules cams, chemokine signaling pathway, complement and coagulation cascades, T cell receptor signaling pathway, adipocytokine signaling pathway, aldosterone regulated sodium reabsorption, glycosaminoglycan biosynthesis heparan sulfate and insulin signaling pathway. Those may bring novel insights into the potential underlying mechanism of SNAP25 in colon cancer. In addition, this paper reveals the significant correlations between SNAP25 and lymphocytes (NK, macrophage, mast cell, NKT), which indicates the potential association of tumor microenvironment and SNAP25. Recently, a lot of attention has been paid to tumor microenvironment and immune evasion for future diagnoses and treatments of malignant tumors. Immunotherapies have been revolutionizing tumor treatment, although immunological response in different patients is heterogeneous. NK cells participate in tumor immunosurveillance, and are one of the most promising therapies for various types of cancer. However, NK cell populations may shape with altered reactivity in malignant tumors (*Hofer & Koehl, 2017*). In colorectal cancer, a high level of mast cells was confirmed with poor survival, and the density of innate immune cells (macrophages, mast cells, neutrophils, and immature dendritic cells) increased with tumor stage (*Koi & Carethers, 2017*). Therefore, SNAP25 is extremely closely associated with various types of TILs, and has the potential to serve as a prognosis biomarker and an immunotherapeutic target for colon cancer.

However, our study presents a number of limitations. One major limitation is that the present research was mainly based on previous data from TCGA and GEO, therefore, future investigations in vivo and vitro are needed to investigate the effect of SNAP25 in colon cancer. A second issue is that since the integrated bioinformatics analysis was focus on immune microenvironment for colon cancer, and the DEGs were identified based on immune and stromal scores, there was no validation for these DEGs based on control samples. Finally, we are lacking of the specimens of colon cancer and adjacent normal tissues. So, we only validate the expressions of AQP4 and SNAP25 in 20 pairs of tissues by qRT-PCR.

## CONCLUSIONS

In summary, SNAP25 is a microenvironment-related and immune-related gene that can predict poor outcomes in colon cancer. Bioinformatic analysis suggests that SNAP25 is

involved in cancer-related signaling pathway, immunity and metabolism processes, which may provide a new target for investigating the underling mechanism of colon cancer.

## ACKNOWLEDGEMENTS

The authors would like to thank Qingqing Xia (Central laboratory of Taizhou First People's Hospital) for her technical assistance in qRT-PCR.

### Funding

This work was supported by one grant from Health and Family planning Commission of Zhejiang Province (2020KY1045) and one grant from Taizhou Science and Technology Bureau in Zhejiang Province (1901ky62). The funders had no role in study design, data collection and analysis, decision to publish, or preparation of the manuscript.

### Grant Disclosures

The following grant information was disclosed by the authors:
Health and Family planning Commission of Zhejiang Province: 2020KY1045.
Taizhou Science and Technology Bureau in Zhejiang Province: 1901ky62.

### Competing Interests

The authors declare there are no competing interests.

### Author Contributions

- Jinyan Zou conceived and designed the experiments, analyzed the data, prepared figures and/or tables, and approved the final draft.
- Darong Duan performed the experiments, prepared figures and/or tables, and approved the final draft.
- Changfa Yu performed the experiments, authored or reviewed drafts of the paper, and approved the final draft.
- Jie Pan and Jinwei Xia analyzed the data, prepared figures and/or tables, and approved the final draft.
- Zaixing Yang conceived and designed the experiments, authored or reviewed drafts of the paper, and approved the final draft.
- Shasha Cai conceived and designed the experiments, performed the experiments, authored or reviewed drafts of the paper, and approved the final draft.

### Human Ethics

The following information was supplied relating to ethical approvals (i.e., approving body and any reference numbers):

The Institutional Medical Ethics Review Board of Taizhou first people's hospital in Zhejiang Province approved the study (Ethical Application Ref: 2019-KY009-03).

## Data Availability

The raw measurements are available in the Supplemental Files.

## Supplemental Information

Supplemental information for this article can be found online at http://dx.doi.org/10.7717/peerj.10142#supplemental-information.

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
