# Peer review of "Mining the potential prognostic value of synaptosomal-associated protein 25 (SNAP25) in colon cancer based on stromal-immune score"

_PeerJ, doi:10.7717/peerj.10142_

## Round 0.1 · original submission · Major Revisions

Please address all the critiques of the reviewers and revise your manuscript accordingly.

Reviewer 1 ·

Basic reporting

The current manuscript described bioinformatic analysis of the correlation of several protein expression with prediction of colon cancer. However, actual experiment to confirm the current study is missing, which greatly reduced the power of the study. The author should find some stained slides or perform WB or qPCR to confirm the current findings.

Experimental design

The experimental design is logic, but no actual experiment was conducted, which makes it scientifically weak.

Validity of the findings

The current finding is valid only based on bioinformatic study, actual experiment should be conducted to confirm the findings.

Additional comments

The current manuscript described bioinformatic analysis of the correlation of several protein expression with prediction of colon cancer. However, actual experiment to confirm the current study is missing, which greatly reduced the power of the study. The author should find some stained slides or perform WB or qPCR to confirm the current findings.

Reviewer 2 ·

Basic reporting

Zou et. al have attempted to provide the potential prognostic value of SNAP25 in colon cancer based. The paper though was well written need to be checked for grammatical errors and spell checks.

line 89

1) Lot of tools and software are not referenced in peerj format, change them to the required format.

Eg:
FunRich 3.1.3 (http://www.funrich.org/) can be reference with https://pubmed.ncbi.nlm.nih.gov/25921073/

3) Relationship between TILs and Survival should be outlined more to drive the point of why correlation between TILs and gene is important.


4) Conclusion should also include with limitation of the methods used.

Experimental design

1) Database and estimation of stromal and immune scores - lacks information about the data such as filters to obtain specific data, you should shift the description of the data from this subheading "The relationships between stromal/immune score and clinical features" to here

2) GEO data- why was only two GSE series was chosen? Also it would be good to show all the 15 genes survival outcomes of hub genes thus your selection of the two genes is justified

3) There is an information gap of how the RNASeq data was normalized?

4) why was the DEG from RNAseq (using counts or RPKM) values was also not considered to see how the 15 hub gene are expressed?

5) Also it would be good to instead information on the control samples that shows the DEGs are really DEGs in normal.

6) GSEA could have been done on all 15 hub gene, thus APQ4 and SNAP25 are prominent in survival.

Validity of the findings

1) The manuscripts lacks the clarity of how only APQ4 and SNAP25 are best in survival outcomes - compared to other 13?

2) The transition from two gene APQ4 and SNAP25 to only SNAP25 focus is unclear? The GSEA was done on both, why was APQ4 not investigated further?

3) The usage of stormal/immune scores for 430 colon cancer patients is concluded with no statistically significant difference. This contradicts the usage of these score and the dataset? (line 145-146)

·

Basic reporting

The manuscript is well written with clear and unambiguous language. The figures are well illustrated with clear descriptions.

There are minor errors to correct:
- Line 50: please remove the 2020 in front of the literature citation
- Line 89: please change flod to fold
- Line 158-163: please include the long form of CC, MF, BP (cellular component, molecular function, biological process/ biological pathway) because this is the first time the short form is used in the main text

Experimental design

The experiment is well designed and performed.

Validity of the findings

The findings and conclusions justify the results shown, and they are statistically sound.

One suggestion to improve data interpretation:
- Figure 5: it is not explained what the dotted blue/ red lines are. Are they the standard deviation of the Kaplan-Meier plot? Please indicate it on the Figure description or the main text.

---

## Round 0.2 · accepted · Accept

Thank you for your clear response to the comments of the reviewers. All critiques were adequately addressed and the manuscript was revised accordingly. I am satisfied with your answers and revisions and happy to accept this work.